# In a Large Healthcare System in the Bronx, Teleretinal Triaging Was Found to Increase Screening and Healthcare Access for an Underserved Population with a High Incidence of T2DM and Retinopathy

**DOI:** 10.3390/ijerph20075349

**Published:** 2023-03-31

**Authors:** Kevin Dahlan, Pamela Suman, David Rubaltelli, Anurag Shrivastava, Roy Chuck, Umar Mian

**Affiliations:** 1Stony Brook Department of Ophthalmology, Stony Brook Medicine, Stony Brook, NY 11794, USA; 2Division of Infectious Disease, Department of Vaccine Center, NYU Langone Health Medical Center, New York, NY 10016, USA; 3Department of Ophthalmology and Visual Sciences, Montefiore Medical Center, Albert Einstein College of Medicine, Bronx, NY 10467, USA; 4Department of Pediatrics, Montefiore Medical Center, Albert Einstein College of Medicine, Bronx, NY 10467, USA

**Keywords:** proliferative diabetic retinopathy, diabetic macular edema, telemedicine, teleretinal, fundus photography, diabetic retinopathy screening

## Abstract

The early treatment of diabetic retinopathy (DR) prevents vision-threatening proliferative retinopathy (PDR) and macular edema (DME). Our study evaluates telemedicine (teleretinal) screening for DR in an inner-city healthcare network with a high ethnic diversity and disease burden. Fundus photographs were obtained and graded in a centralized reading center between 2014 and 2016. Patients with positive screenings were referred to a retina specialist. An analysis of sensitivity and specificity and a subgroup analysis of prevalence, disease severity, and follow-up adherence were conducted. In 2251 patients, the ‘1-year’ and ‘Overall’ follow-ups were 35.1% and 54.8%, respectively. Severe grading, male gender, and age were associated with better follow-up compliance. The DR, PDR, and DME prevalence was 24.9%, 4.1%, and 5.9%, respectively, and was significantly associated with HbA1c. The sensitivity and specificity for DR, PDR, and DME were 70% and 87%, 87% and 75%, and 37% and 95%, respectively. No prevalence differences were noted between ethnicities. Annual diabetic eye exam adherence increased from 55% to 85% during the study period. Teleretinal triaging is sensitive and specific for DR and improved diabetic eye exam compliance for underserved populations when integrated into large healthcare networks. The adherence to follow-up recommendations was better among older patients and among those with more severe retinopathy.

## 1. Introduction

Diabetic retinopathy (DR) is one of the most common complications of diabetes, with a prevalence of 28.5% in the US diabetic population [1,2]. It is the leading cause of new cases of blindness in patients aged 20 to 74, accounting for about 15–17% of the cases in the United States and Europe [1,3]. The resulting vision loss leads to personal and financial costs which contribute to a significant burden of healthcare spending, with the estimated spending being over USD 500 million dollars annually [4]. It is well established that the early detection and treatment of DR decreases visual complications when implemented in large health systems [5,6]. Despite the current recommendation for at least one annual ophthalmology examination for diabetic patients, only half of these patients adhere with the recommendations [7]. Disparities in healthcare access for ethnic minorities lead to differences in both the screening rates and severity of retinopathy and highlight the importance of access to effective screening and treatment [8,9,10,11]. Given the limitations in screening for diabetic retinopathy, it is critical to identify and validate systems that can be developed to effectively not only diagnose disease, but furthermore to expedite intervention. 

Remote telemedicine screening methods have been validated in multiple studies to be effective and relatively inexpensive methods for evaluating diabetic retinopathy, with both a high sensitivity and specificity for detecting retinopathy [8]. They have been used effectively to triage patients for ophthalmology referral based on disease screening grade severity [12]. Telemedicine screening has been demonstrated to be widely implementable and preferred by patients over routine clinical exam., In one analysis, it was demonstrated to double the overall screening rates for traditionally underserved ethnic subgroups by improving remote access [13]. While two-dimensional teleretinal screening is sensitive to DR, there are important limitations to consider. For example, fundus photography relies on the presence of macular exudates as a surrogate for macular edema instead of objective measurement of macular thickness. Although this leads to an overall lower sensitivity for detecting potentially sight threatening macular edema, [14] these systems regardless have extraordinary utility to increase screening rates for DR despite the limitations.

The Montefiore Health System consists of multiple hospitals and ambulatory care centers in the Bronx, New York. This diverse patient population includes 1.3 million people and consists of a large proportion of patients of a lower socioeconomic status, with 80% of patients using government payer healthcare insurance. Median household income in 2019 was USD 43,540, which is 40% less than the citywide median household income (USD 72,930), with a higher poverty rate of 26.4% compared to 16% citywide [15]. 

Our patient population is known to be vulnerable to debilitating vision loss, and has a concomitantly high prevalence of type 2 diabetes (T2DM) and relatively low compliance with preventive screening [16]. In response, the Montefiore Health System piloted a primary-care-based telemedicine initiative to triage and refer patients at risk of vision loss from diabetic retinopathy to appropriate subspeciality care. This study details the sensitivity and specificity of this telemedicine screening program for detecting diabetic retinopathy (DR), proliferative diabetic retinopathy (PDR), and macular edema (DME), the prevalence of DR, PDR, and DME in the population, and rates of follow-up for patients in the program. 

## 2. Materials and Methods

The protocol for this study was approved by the Albert Einstein College of Medicine institutional review board and all methods performed were in accordance with the relevant guidelines and regulations.

### 2.1. Study Population

Adult diabetic patients (>18 years old) were recruited from eight primary care clinics within the Montefiore Medical Center’s community network between June 2014 and July 2016, and followed for adherence to recommendations until 2018. The diagnosis of diabetes was defined by use of anti-glycemic medications and/or elevated HbA1C [17]. Patients were offered diabetic telemedicine photography in their primary care physician’s clinic on the same day as their routine visit. Patients who were unable to be imaged (i.e., media opacification, poor view, etc.) were excluded and referred to subspecialty care at the PCPs discretion. 

### 2.2. 1-Field Fundus Photography

A non-mydriatic camera was used (Nidek AFC-230 [NIDEK Inc, San Jose, CA, USA] or Ioptics Easyscan SLO [Cassini, Burlington MA, USA]) to take 45-degree macula-centered fundus photos in each eye. Trained technicians (Healpros LLC (Atlanta, GA, USA) were sent to each primary care office to acquire retinal photographs, and subsequently upload the files to our electronic medical record (Epic Systems, Verona, WI, USA) for standardized interpretation by a reading center staffed by one of five board-certified Montefiore ophthalmologists. 

### 2.3. Image Grading and Referrals

The quality of image was graded to be fully, partially, or not assessable by the reading center. All screened patients received an overall diabetic retinopathy severity grade (using the NSC-UK grading criteria) determined by the worse grade between the two eyes [18]. In cases where one eye had an ‘indeterminate grade’, the screening was defined as positive if the contralateral eye had a grade of ‘background retinopathy’ or worse. Patients with one eye graded as ‘no retinopathy’ and the contralateral as ‘indeterminate’ were considered to be ‘indeterminate’. Follow-up times were based on the patient’s diabetic or non-diabetic findings and were noted in the report.

Once images were officially graded, the primary care team alerted patients of the results, and a call center was tasked to schedule the patient (up to 3 attempts) with a Montefiore ophthalmologist as per the screening recommendation. Patients graded as ‘indeterminate’ were suggested to follow-up as if severe disease was detected to limit the impact of false negative examinations. A comprehensive eye exam was performed at the follow-up clinic visits. The time interval between the screening and the first follow-up was recorded.

### 2.4. Demographics

A retrospective chart review was performed to collect demographic information, which included the patient’s age, gender, and self-reported race/ethnicity (non-Hispanic black (NHB), non-Hispanic white (NHW), Hispanic/Latino/Spanish (HLS), and other (Asian, Native American, ‘not applicable’, Other)). The HbA1c available at the time of screening and details of follow-up clinic appointments with ophthalmologists were also collected. A single health insurance provider’s annual diabetic eye exam adherence data were used as a surrogate for changes in compliance over the entire hospital network for the study period.

### 2.5. Statistical Methods

Statistical analyses were conducted using SAS version 9.4 software. To examine the difference between race/ethnicity in terms of diabetic retinopathy, proliferative retinopathy, 1-year follow-up (1YFU), and overall follow-up (OFU), a logistic regression model adjusted for age, gender, and A1C was used. Analyses involving the follow-up were conducted, controlling for the overall grade of diabetic retinopathy. To test whether there is any difference in intervals between the screening and overall follow-up among different ethnicity groups, a Kaplan–Meier curve and log rank tests were used. The Cox proportional hazard model was also used, adjusting for age, gender, A1C, and the overall grade of diabetic retinopathy.

Odds ratios and 95% confidence intervals were calculated for each comparison. Severity of retinopathy was reported in patients with at least one gradable image based on the level of disease in the worse eye. To measure sensitivity and specificity, dilated fundus examination was compared to teleretinal images. A positive screen was defined as a graded image of background retinopathy or more severe in at least one eye. Patients graded as ‘indeterminate’ were excluded in the sensitivity and specificity analysis, as well as statistics involving the screening grade severity. These patients were included for the analysis of follow-up rates and demographic information, however.

## 3. Results

A total of 2251 patients were included in the analyses. The average age of the study participants was 59.9 years of age (20–96 years old). A total of 60.6% of patients were female, with three patients declining to list their gender. The racial/ethnic makeup of the population was HLS (45%), NHB (36%), and NHW (4%), and 16% were listed as ‘other’. The average A1c of the population was 8.15% (3.9–17.4%). NHW patients had an average A1c of 7.62%, which was lower than that of HLS and NHB patients, with A1c values of 8.16% (*p* = 0.03) and 8.11% (*p* = 0.058), respectively (Table 1).

Of the 4497 images analyzed, 87.12%, 11.45%, and 1.42% of images were graded as fully, partially, or not assessable, respectively (five patients were only able to have an image of one eye). Overall, 71.4% of patients screened negative and 23.1% of patients screened positive, with 18.9%, 3.38%, and 0.76% receiving grades of background, pre-proliferative, and proliferative diabetic retinopathy, respectively. A total of 5.6% of patients received indeterminate grades.

The overall office follow-up rate for triaged patients was 54.78% by the end of the study (1233 patients), with 35.14% of patients attending follow-up appointments within the first year after teleretinal screening (Table 1). The average time interval for patients between the screening and follow-up was 345 days (4–1393 days). The OFU for patients with background, pre-proliferative, and proliferative retinopathy was 61%, 61%, and 71%, respectively. Patients with more severe grades of retinopathy were more likely to follow up, regardless of the follow-up interval (OR = 1.42, 95% CI 1.14–1.77, *p* = 0.0019), and within 1 year (OR = 1.60, 95% CI 1.28–2.00, *p* = 0.0003). Age was an independently significant predictor for 1YFU and OFU, with patients 1% more likely to follow up for each year of age (1YFU OR = 1.01, 95% CI 1.01–1.02, *p* = 0.0006; OFU OR = 1.01, 95% CI 1.00–1.02, *p* = 0.0204). There was no significant relationship between race/ethnicity and the follow-up rate when controlling for age and diabetic retinopathy screening grade severity (Table 2). During the program, the annual adherence to standard-of-care yearly diabetic eye exams for diabetic patients increased every year. An increase of 30% was noted from the start to the end of the study (adherence of 55% in 2014 to 85% in 2016).

For the OFU patients, there was a 24.9% prevalence of DR, a 4.1% prevalence of PDR, and a 5.9% prevalence of DME (Table 3). A1c was independently associated with both DR (OR = 1.29, 95% CI = 1.21–1.37, *p* < 0.0001) and PDR (OR = 1.33, 95% CI 1.20–1.48, *p* < 0.0001) diagnosed in the clinical exam. HLS and NHB patients had a DR prevalence of 26% and 25%, respectively, while NHW patients had a prevalence of 12%. For PDR, HLS and NHB patients had a prevalence of 4% and 3%, respectively, while NHW patients had a rate of 2%. When controlling for age and A1c, however, there was no significant difference in DR or PDR prevalence between any of the races/ethnicities (Table 4). The sensitivity and specificity of the screening test, when compared with the results from the clinical exam, were 70% and 87% for DR, 87% and 75% for PDR, and 37% and 94% for DME, respectively.

## 4. Discussion

In a large healthcare system in the Bronx, the implementation of teleretinal triaging was found to increase screening and healthcare access for an underserved population with a high incidence of T2DM and retinopathy. The Medicare quality rating for diabetic eye examinations in the institution increased from one star to five stars by the end of the study period. As discussed in a study by Muqri et al. on the cost-effectiveness of this triaging program, the downstream revenue totaled USD 280,000 for the health system, with 14.66 quality-adjusted life-years gained during the program due to the early treatment of the disease [19]. When our screening methodology was compared with other telemedicine programs that used one-field fundus photography, it had a sufficient yet lower sensitivity and a higher specificity for DR when compared to dilated ophthalmoscopy [14,20]. The poor image quality may have contributed to the decreased sensitivity, as ~12% of pictures were deemed only partially readable. Lee et al. found that the sensitivity increased to 100% and the specificity increased to 77% when using nonmydriatic three-field fundus photography, suggesting that this method may have the highest sensitivity for screening. Additionally, they also noted a higher number of ungradable readings when using single-field photography compared to using multiple fields [20]. As expected for DME, our study sensitivity was similarly low compared to that of other studies and stems from the inability to detect foveal thickness on fundus photography and from using macular exudates as a surrogate for DME [14]. 

The study population, when compared to the standard patient population of the Bronx, had a lower percentage of NHW (4% vs. 9%) and HLS patients (45% vs. 56%) and a higher percentage of NHB patients (36% vs. 29%) [21]. The program was able to improve the delivery of initial primary screening to underserved patients. However, the follow-up rate in the clinic following screening was similarly low in relation to the follow-up rates for other urban populations [22]. For instance, similar studies in diverse urban populations reported a follow-up rate over 3 years of 52.9% after the ophthalmology referral, which is comparable to the overall follow-up rate in our study of 55% over 4 years [14]. Interestingly, screening grade severity was most likely to predict a higher follow-up, suggesting that patients may perceive continued follow-up as important if they have worse disease. Supporting this idea, other studies have shown that patients have lower follow-up if they perceive that they are disease-free [23]. Unlike other studies that demonstrated racial disparities in follow-up rates, our study did not demonstrate such an association [8,9,10,11]. This, however, may be due to the small number of patients in this population self-identifying as NHW. Additionally, follow-up was only recorded for patients seeing ophthalmologists within the Montefiore system. Patient visits with ophthalmologists outside of the system could not be recorded, likely underestimating follow-up rates. 

The proportion of diabetic patients with DR and PDR in this study was similar to the prevalence reported nationwide [1]. Poor glycemic control has been shown to increase the prevalence of DR and PDR [1,24,25]. In our study, a higher A1c was also a predictive independent factor for DR and PDR in our patient population. The A1c of the study population is representative of the Bronx population, with a similar average of 8.2%, higher than the average A1c values of other boroughs in NYC [26]. There are conflicting reports of associations between DR and race/ethnicity, with some demonstrating a positive association [1,9,27,28], whereas others found no association [28,29]. Similarly, our study demonstrates no independent relationship between race/ethnicity and the prevalence of DR and PDR when controlling for A1c and age. 

This study demonstrated that implementing teleretinal triaging initiative in a large urban healthcare system was able to improve adherence to annual examination in a population of underserved diabetic patients. The program increased the availability of primary screening for thousands of diabetic patients in the Bronx and offered sight-saving access to treatment for affected patients. The relatively low adherence to follow-up after the initial triage visit is likely the result of a multitude of social determinants of health, and represents a challenge that needs to be addressed in order to improve access to care to our most vulnerable patients. 

## 5. Conclusions

An urban telemedicine initiative utilizing a centralized reading center and non-mydriatic single-field fundus photography performed in a primary care setting was successfully able to detect DR and, in particular, PDR with a high sensitivity and specificity. The subsequent adherence to follow-up specialty care improved over the course of the program, and it was significantly correlated with older age, male gender, and a higher grading of DR.

## Figures and Tables

**Table 1 ijerph-20-05349-t001:** Demographic Information and Baseline Characteristics.

Race/Ethnicity	N (%)	M:F Ratio	Age (Range)	A1c (Range)	1-Year Follow-Up (%)	Total Follow-Up (%)
Non-Hispanic White	81 (4)	2.00	62.5 (31–84)	7.62 (5.0–14.7)	28 (35)	41 (51)
Non-Hispanic Black	800 (36)	0.50	59.5 (21–94)	8.11 (3.9–17.4)	287 (36)	440 (55)
Hispanic	1016 (45)	0.63	60.4 (21–92)	8.16 (4.6–17.4)	346 (34)	539 (53)
Other	354 (16)	0.96	58.7 (20–96)	8.34 (4.9–17.4)	131 (37)	213 (60)
Total	2251	0.65	59.9 (20–96)	8.15 (3.9–17.4)	792 (35)	1233 (55)

Demographic information of the study population. Percentages describe the percentage of patients for each race/ethnicity category.

**Table 2 ijerph-20-05349-t002:** Results of the Linear Regression of Predictors for Follow-ups in the Clinic.

Follow-Up within 1 Year			Any Follow-Up		
Factor	Odds Ratio (95 % CI)	*p* Value	Factor	Odds Ratio (95% CI)	*p* Value
Non-Hispanic Black (287) *	0.99 (0.60–1.61)	0.96	Non-Hispanic Black (440) *	1.09 (0.68–1.74)	0.72
Hispanic (346) *	1.10 (0.67–1.79)	0.7	Hispanic (539) *	0.96 (0.61–1.53)	0.88
Age	1.01 (1.01–1.02)	**0.0006**	Age	1.01 (1.00–1.02)	**0.02**
Gender ^†^	1.11 (0.92–1.33)	0.27	Gender ^†^	1.244 (1.05–1.48)	**0.01**
A1c	1.02 (0.97–1.06)	0.47	A1c	1.01 (0.97–1.05)	0.51
DR Screening Grade ^‡^	1.60 (1.28–2.00)	**<0.0001**	DR Screening Grade ^‡^	1.42 (1.14–1.77)	**0.0019**

Results of logistic regression for factors affecting follow-up. * Race/ethnic categories were compared to NHW patients. ^†^ Gender compared males to females. ^‡^ Results compared patients with no retinopathy and patients with background retinopathy. **Bolded** values are statistically significant < 0.05.

**Table 3 ijerph-20-05349-t003:** Prevalence of DR, PDR, and DME in patients who followed up in the clinic.

Race/Ethnicity (N)	A1c (Range)	DR (%)	PDR (%)	DME (%)
Non-Hispanic White (41)	7.62 (5.0–14.7)	5 (12)	1 (2)	1 (2)
Non-Hispanic Black (440)	8.11 (3.9–17.4)	110 (25)	14 (3)	22 (5)
Hispanic (539)	8.16 (4.6–17.4)	141 (26)	24 (4)	35 (6)
Other (213)	8.34 (4.9–17.4)	40 (19)	8 (4)	15 (7)
Total (1233)	8.15 (3.9–17.4)	296 (24)	47 (4)	73 (6)

Prevalence of DR, PDR, and DME found in the follow-up exam in the clinic. Percentages describe the proportion of patients with each disease for each race/ethnicity category.

**Table 4 ijerph-20-05349-t004:** Results of the linear regression of risk factors for DR and PDR prevalence.

Diabetic Retinopathy			Proliferative Diabetic Retinopathy		
Factor	Odds Ratio (95% CI)	*p* Value	Factor	Odds Ratio (95% CI)	*p* Value
Black (440) *	2.187 (0.818–5.848)	0.119	Black (440) *	0.98 (0.12–7.84)	0.98
Hispanic (539) *	0.422 (0.159–1.122)	0.084	Hispanic (539) *	0.71 (0.09–5.51)	0.74
Age	1.007 (0.995–1.019)	0.239	Age	1.00 (0.98–1.03)	0.84
Gender ^†^	0.867 (0.653–1.149)	0.32	Gender ^†^	1.20 (0.64–2.27)	0.57
A1c	1.29 (1.214–1.37)	**<0.0001**	A1c	1.33 (1.20–1.48)	**<0.0001**

Statistics regarding risk factors for DR and PDR prevalence. * Race/ethnic categories were compared to non-Hispanic White patients. ^†^ Gender compared males to females. **Bolded** values are statistically significant < 0.05.

## Data Availability

The datasets used and/or analyzed during the current study are available from the corresponding author on reasonable request.

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
