# Peer review of "In a Large Healthcare System in the Bronx, Teleretinal Triaging Was Found to Increase Screening and Healthcare Access for an Underserved Population with a High Incidence of T2DM and Retinopathy"

_ijerph, 2023, doi:10.3390/ijerph20075349_

Round 1

Reviewer 1 Report

Line 22 -it is unclear which gender has a better compliance (not even Table 2 answers to that question)

Line 79 – “greater than 90% of patients have T2DM” – this is a general statement and may be used in the introduction. I don’t think it has a place in the Materials and methods paragraph. In the Results section you may state what is the percentage of T2DM in your study population.

Table 2 is stating that patients with background retinopathy are more likely to adhere to follow-up than those without retinopathy, and lines 209-211 are stating that screening grade severity predicts higher follow-up. It would be interesting to know what were the rates of follow-up for patients screened with background, pre-proliferative and proliferative retinopathy.

Author Response

Cover Letter

The authors would like to personally thank both reviewers for their esteemed suggestions to improve the quality of our paper. For convenience, I included the question/suggestion of each reviewer in italics with the corresponding changes in the article to address these suggestions.

Review Report #1:

  • Line 22 -it is unclear which gender has a better compliance (not even Table 2 answers to that question)
    • Included change clarifying that men had improved follow-up compliance – Line 22
  • Line 79 – “greater than 90% of patients have T2DM” – this is a general statement and may be used in the introduction. I don’t think it has a place in the Materials and methods paragraph.
    • Removed reference to the T2DM prevalence statistic from the materials and methods section – Line 86-87
  • Table 2 is stating that patients with background retinopathy are more likely to adhere to follow-up than those without retinopathy, and lines 209-211 are stating that screening grade severity predicts higher follow-up. It would be interesting to know what were the rates of follow-up for patients screened with background, pre-proliferative and proliferative retinopathy.
    • Overall follow-up for patients with background, pre-proliferative and proliferative retinopathy was 61%, 61%, and 71% respectively – Lines 163-164

Reviewer 2 Report

The article will gain interest if the authors would start with a clear description of the problems they encounter in their longing to treat their patients right.

Maybe this is partly described in the reference number 15, however for the reader of this article not clear. 

So, how big is the population they want to serve? Where is it located exactly?What is the socio-economic background? How is the overall health and how is that measured? How many primary clinics are present in the area? How is the acces to these clinics? Are the clinics understaffed? Do the clinicians suffer of language barriers? Why do people come/ do not come? If they do not visit their follow up appointment, why not? Are they reminded by messages on their cell phone? Is it because of money?

How much people are underserved, how do you know and what is underserved? Which options are available to reach these people?

Why telemedicine and how are the results from tele medicine communicated to the patients? Could it be done by smart-phone?

I also miss the recommendations for life-style changement or any description on how this is communicated/ brought to the patients.

Because diabetes is also such an high financial burden, a well described definition of the magnitude the existing problems and consequences of late or no treatment, could hopefully invite governments/ NGO's/ health insurance compagnies to invest financially in prevention (a.o.by telemedicine)

Author Response

The authors would like to personally thank both reviewers for their esteemed suggestions to improve the quality of our paper. For convenience, I included the question/suggestion in italics with the corresponding changes in the article to address these suggestions.

Reviewer Report #2:

  • So, how big is the population they want to serve? Where is it located exactly?
    • The study took place in the Bronx, New York. This diverse patient population includes 1.3 million people - lines 62-63
  • What is the socio-economic background? How is the overall health and how is that measured. How much people are underserved, how do you know and what is underserved? Which options are available to reach these people?
    • Prevalence of diabetes and socioeconomic status was taken from prior studies at Montefiore with 80% of patients using government payer healthcare insurance. Median household income in 2019 was $43,540 which is 40% less than the citywide median household income ($72,930) with a higher poverty rate of 26.4% compared to 16% citywide. – lines 65-67
  • How many primary clinics are present in the area? How is the access to these clinics? Are the clinics understaffed? Do the clinicians suffer of language barriers?
    • This study worked in concert with 8 primary care physician offices in the community to increase screening compliance for their patients. Patients made appointments with their primary physicians and underwent telemedicine triaging at their physician’s office. Each Montefiore Health System office can access translation services over the phone to reduce language barriers.
  • Why telemedicine and how are the results from tele medicine communicated to the patients? Could it be done by smartphone? I also miss the recommendations for lifestyle changement or any description on how this is communicated/ brought to the patients.
    • Primary care physicians identified patients with diabetes for telemedicine screening at their clinics without having to make an additional appointment with the ophthalmologist to increase access to eye care for these patients. Results and recommendations were given to patients through their primary care physician, and they were scheduled for follow-up through the Montefiore call center. Lifestyle modification discussions were provided by the primary care providers and nutritionists within the Montefiore system – lines 111-116
  • Why do people come/ do not come? If they do not visit their follow up appointment, why not? Are they reminded by messages on their cell phone? Is it because of money?
    • While an important and interesting question, the reasons why patients did not make follow-up appointments was not determined from patients who were lost to follow-up and should be the subject of future studies to increase compliance – lines 240-244
  • Because diabetes is also such an high financial burden, a well described definition of the magnitude the existing problems and consequences of late or no treatment, could hopefully invite governments/ NGO's/ health insurance compagnies to invest financially in prevention (a.o.by telemedicine)
    • This triaging program was cost-effective, generating downstream revenue totaling $280,000 for the health system with an estimated 14.66 quality adjusted life-years gained during the program. – lines 200-202